# Altering the spectroscopy, electronic structure, and bonding of organometallic curium(III) upon coordination of 4,4′−bipyridine

**Brian N. Long** [1], **María J. Beltrán-Leíva**[1], **Joseph M. Sperling**[1], **Todd N. Poe** [1], **Cristian Celis-Barros** [1,2] ✉ & **Thomas E. Albrecht-Schönzart** [1,2] ✉

Structural and electronic characterization of $(Cp'_3Cm)_2(\mu{-}4,4'{-}bpy)$ (Cp′ = trimethylsilylcyclopentadienyl, 4,4′−bpy = 4,4′−bipyridine) is reported and provides a rare example of curium−carbon bonding. $Cp'_3Cm$ displays unexpectedly low energy emission that is quenched upon coordination by 4,4′−bipyridine. Electronic structure calculations on $Cp'_3Cm$ and $(Cp'_3Cm)_2(\mu{-}4,4'{-}bpy)$ rule out significant differences in the emissive state, rendering 4,4′−bipyridine as the primary quenching agent. Comparisons of $(Cp'_3Cm)_2(\mu{-}4,4'{-}bpy)$ with its samarium and gadolinium analogues reveal atypical bonding patterns and electronic features that offer insights into bonding between carbon with $f$-block metal ions. Here we show the structural characterization of a curium−carbon bond, in addition to the unique electronic properties never before observed in a curium compound.

Discovered by Seaborg, James, and Ghiorso in 1944 via $\alpha$-particle bombardment of $^{239}$Pu, curium ($Z = 96$) is one of the heaviest elements available in quantities suitable for traditional synthetic chemistry[1]. It is most stable in the +3 oxidation state, and possesses a $[Rn]5f^7$electron configuration[1]. Its half-filled shell creates increased stability with respect to other $5f^n$ configurations, and is often associated with an expected decrease in $f$-electron contributions to bonding, as well as high resistance to changes in oxidation state[2–4]. Additionally, similarities in ionic radii and trivalent stability lead to challenges in separating curium from americium and the lanthanides. This separation is an essential component of recycling used nuclear fuel owing to curium's significant contribution to the radiotoxicity of nuclear waste[5–10].

Despite the distinct electronic properties of the wide variety $Cm^{3+}$ compounds that have been prepared to date, no single-crystal structural characterization of a complex containing a Cm−C bond has been reported[6,11–20]. Examination of this interaction could provide insights into methods for engaging the frontier orbitals of curium and other actinides in forming partially covalent bonds with selected ligands.

This could allows us to gain some control over the electronic structure of these complex elements[21–30]. For example, it has been shown that the engagement of $5f$ orbitals in Cm-ligand bonds can be increased by using soft-donor ligands that can be further enhanced by applying mechanical pressure[3].

Quantum mechanical evaluation of a recently reported Am(III) cyclopentadienyl (Cp) complex showed that a variety of metal frontier orbitals were mixing with Cp′ ligand orbitals to create partially covalent bonds, but also revealed a surprising degree of ionicity in the Am−N interactions with 4,4′-bipyridine in the same complex[31,32]. However, complexes of this type remain rare largely due to low available quantities of these isotopes (reactions are completed with <5 mg of metal content), the need for specialized research faciltities, and their exceptional air and moisture sensitivity[24,30–33]. Owing to these difficulties, syntheses allowing the structural characterization of An−C (An = Pu, Am, Cf ) have only recently become accessible and can be applied to curium[24,30–34]. These synthetic challenges also necessitate the use of lanthanide analogs that possess similar ionic radii and/or

[1]Department of Chemistry and Biochemistry, Florida State University, 95 Chieftan Way, Tallahassee, FL 32306, USA. [2]Department of Chemistry and Nuclear Science & Engineering Center, Colorado School of Mines, Golden, CO 80401, USA. ✉e-mail: ccelisbarros@mines.edu; talbrechtschoenzart@gmail.com

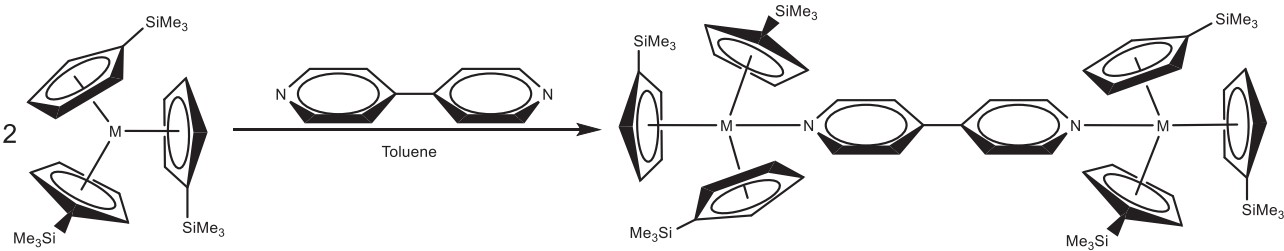

**Fig. 1 | Chemdraw of the reaction between Cp′₃M and 4,4′−bpy.** Addition of 4,4′−bpy to two equivalents of Cp′₃M (M = Sm, Gd, Cm) yields (Cp′₃M)₂(μ−4,4′−bpy). Single crystals are obtained by cooling hot toluene to room temperature.

electron configurations for optimizing the chemistry and for providing benchmarks for comparisons with the 5*f* series. Traditionally, lanthanide and early actinide organometallic chemistry has lead to the characterization of a suite of new oxidation states, electron configurations, and splitting of *f*-*f* transitions spectroscopically, greatly contributing to our fundamental understanding of these elements[24–27,29]. Utilizing these systems for the mid-actinides has historically provided an excellent basis for comparison to lanthanides and early-actinide counterparts on a fundamental level, leading to future studies in new oxidation states, covalency, and electronic properties of curium[24,30–32].

The synthesis and first single-crystal structural characterization of Cm−C bonding is reported, in addition to its lanthanide analogs with Sm(III) and Gd(III) (Fig. 1). This multinuclear organometallic curium complex, (Cp′₃Cm)₂(μ−4,4′−bpy) (Cp′ = trimethylsilylcyclopentadienyl, 4,4′−bpy = 4,4′−bipyridine) (1−Cm), serves as gateway into the field of organometallic curium chemistry and provides a further understanding of soft-donor coordination with curium. Additionally, 1−Cm and its putative Cp′₃Cm precursor present unexpected spectroscopic properties atypical of curium systems. Comparison to lanthanide analogs, (Cp′₃Sm)₂(μ−4,4′−bpy) (1−Sm) and (Cp′₃Gd)₂(μ−4,4′−bpy) (1−Gd), are discussed based on similarities in ionic radii and valence, respectively. This study further enforces the influence of a crowded coordination environment on the degree of covalency in soft-donor transuranic systems and introduces bonding variances between Cm−C bonding from its organometallic lanthanide analogs.

## Results

Addition of 4,4′−bpy to two equivalents of a putative Cp′₃M (M = Sm, Gd, Cm) in toluene yields (Cp′₃M)₂(μ−4,4′−bpy) (Fig. 2). All three molecules crystallized in the *P*1̄ space group and were isomorphous, demonstrating a pseudo-tetrahedral geometry from the 4,4′−bpy and centroids of three Cp′ rings coordinated to a metal center. In each case, two metal centers are bridged by 4,4′−bpy, creating a dinuclear barbell shape with an inversion center in the center of the 4,4′−bpy. Additional synthetic and crystallographic details can be found in the Supplementary Information. Cif files for 1−Sm, 1−Gd, and 1−Cm crystal structures are provided as Supplementary Data 1, Supplementary Data 2, and Supplementary Data 3, respectively.

1−Sm, 1−Gd, and 1−Cm possess M−N distances of 2.626(3) Å, 2.592(3) Å, and 2.5962(16) Å, respectively. Expectedly, 1−Gd possesses a slightly shorter distance than 1−Sm owing to the lanthanide contraction. However, the M−N distance observed in 1−Cm is within error of the 4*f* congener, 1−Gd, but is substantially shorter than its ionic radii-based analog, 1−Sm[35]. The M−N distance presented in 1−Sm is slightly shorter than that of a similar molecule, Cp₃Sm(py) (py = pyridine), 2.656(3) Å[36]. A longer distance would be anticipated in 1−Sm based on the provided steric bulk from the trimethylsilyl groups of Cp′ competing with 4,4′−bpy. However, 1−Gd exhibits a notable longer Gd−N distance in comparison to that reported in Cp₃Gd(NH₃), 2.501(6) Å, due to the significantly smaller size of the coordinated NH₃[37]. As there has been no single-crystal structural characterization of organometallic curium to date, comparison to similar coordination environments proves

challenging. The Cm−N distance of 1−Cm is greater than those reported in Cm(HDPA)₃·H₂O (H₂DPA = 2,6−pyridinedicarboxylic acid) but quite similar to Cm(S₂CNEt₂)₃(N₂C₁₂H₈)[10,38]. The average Cm−N distance of Cm(HDPA)₃ is 2.541(4) Å and 2.550(4) Å for the Λ and Δ enantiomers, respectively, and the reported Cm−N distance Cm(S₂CNEt₂)₃(N₂C₁₂H₈) is 2.601(6) Å[10,38]. Curium sets a trend for these bridged organometallic actinide systems in the trivalent state. Previously reported values of isostructural systems containing uranium[39] and americium[31] possess notably longer M−N bonds than 1−Cm: 2.626(7) Å, 2.618(2) Å, and 2.5962(16) Å respectively, introducing a non-linear trend of decreasing M−N distances across the actinide series. A possible explanation for the greater difference between the Am−N and Cm−N distance is a greater degree of covalency observed in M−N bond of 1−Cm compared to the ionic M−N bond in the americium structure[31]. This trend excludes thorium owing to its low stability in the trivalent state, resulting in the reduction of 4,4′−bpy and oxidation to Th⁴⁺[40]. (Cp′′₃Th)₂(μ−4,4′−bpy) possesses a M−N distance of 2.362(4) Å, significantly shorter than the reported trivalent systems[40].

The M−Cent (Cent = Centroid) bond distances observed in 1−Sm, 1−Gd, and 1−Cm are 2.516 Å (*std.* 0.017 Å), 2.498 Å (*std.* 0.019 Å), and 2.517 Å (*std.* 0.016 Å), respectively. Distorted symmetry and steric competition resulting from the increased coordination number causes a deviation in M−Cent distances compared to their Cp′₃M counterparts (Table S5, S6)[27,28]. A range of M−Cent distances from 2.498(2) Å to 2.538(2) Å is seen in 1−Cm, 2.496(3) Å to 2.537(3) Å in 1−Sm, and 2.477(4) Å to 2.523(4) Å in 1−Gd, presenting a statistically significant variation of about 0.04 Å in all three molecules. 1−Gd exhibits a slightly shorter range than 1−Sm; however, 1−Sm is within error of 1−Cm. A decrease in M−Cent_avg is observed in comparison to isostructural uranium and americium systems[31,39]. The average M−Cent length and range observed in 1−Gd are longer and spread over a broader range compared to the reported M−Cent distance range seen in Cp′₃Gd, 2.434 Å–2.441 Å owing to increased coordination[28]. Similarly, the Cent−M−Cent angles of 1−Sm, 1−Gd, and 1−Cm are lower than the reported Cent−M−Cent angles of Cp′₃M (M = lanthanide), due to the effect of 4,4′−bpy on the coordination environment[27,28]. The observed average Cent−M−Cent angle in 1−Cm is 117.3°. The Cent−M−N angles of 1−Cm range from 93.761° to 103.628°. This considerable difference can be attributed to the bulkiness of the trimethylsilyl groups of Cp′. Additionally, 4,4′−bpy is angled downward at just over 9°, that is likely due to an interaction between hydrogen from the trimethylsilyl groups and nitrogen on the bridge.

Average M−C bond distances of 1−Sm, 1−Gd, and 1−Cm are 2.788 Å (*std.* 0.041 Å), 2.771 Å (*std.* 0.048 Å), and 2.789 Å (*std.* 0.047 Å), respectively. Similar to the M−Cent bonds, 1−Sm possesses longer average M−C distances than 1−Gd, consistent with what is expected from the lanthanide contraction. Additionally, 1−Sm and 1−Cm share similar average M−C distances, within error, owing to nearly identical ionic radii[35]. A significant variation in M−C distances is observed due to ring shifting upon the coordination of 4,4′−bpy. A broad range of M−C distances, between 2.714(4) Å and 2.889(3) Å, is noted in 1−Gd, differing greatly from those reported in Cp′₃Gd, 2.690(2) Å–2.7427(19) Å[28]. This comparison remains consistent between 1−Sm and Cp′₃Sm[27]. 1−Cm

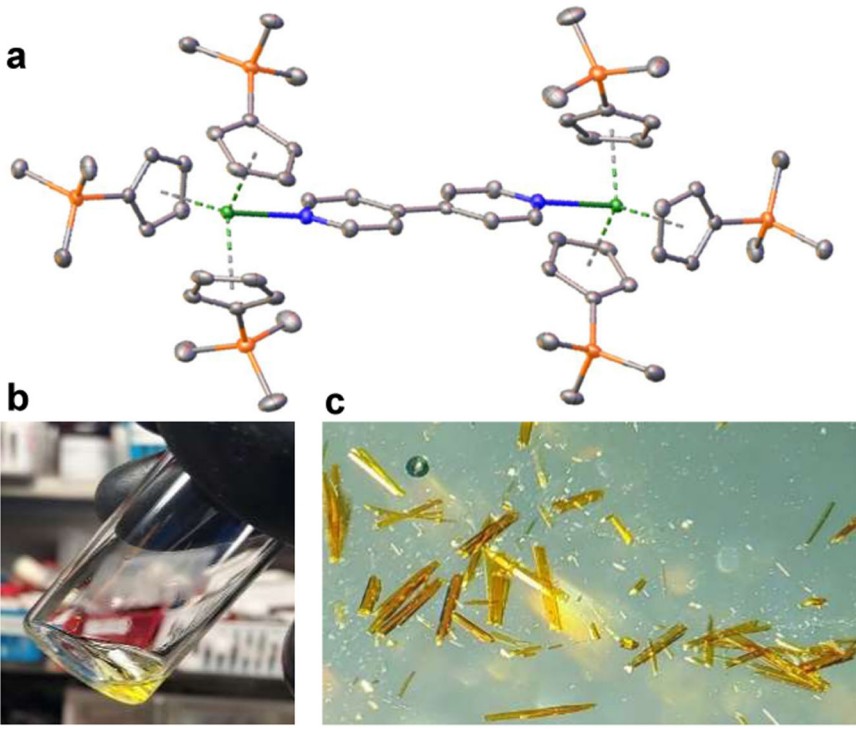

**Fig. 2 | Crystals and structure of 1−Cm. a** Structure of 1−Cm with thermal ellipsoids at 50% probability. Green = Curium, Blue = Nitrogen, Orange = Silicon, Gray = Carbon, and hydrogen have been omitted for clarity purposes. **b** A concentrated solution of 1−Cm in toluene. **c** Crystals of 1−Cm.

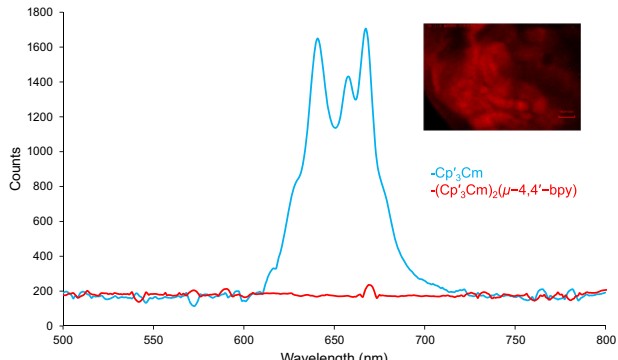

**Fig. 3 | Photoluminescence of putative Cp′₃Cm and 1−Cm.** Photoluminescence of putative Cp′₃Cm microcrystals (blue) compared to 1−Cm (red), excited at 420 nm at −180 °C. Cp′₃Cm showed glowing red upon irradiation at 420 nm. Quenching of emission upon the coordination of 4,4′−bpy is observed.

experiences the same phenomenon, exhibiting M−C distances between 2.728(2) Å and 2.9029(2) Å within the same Cp′⁻ ring. This substantial range indicates that coordination of 4,4′−bpy not only directly impacts M−C distances, but also the angle at which Cp′ coordinates. The average M−C distance in 1−Cm is shorter than those reported in (Cp′₃U)₂(μ−4,4′−bpy)[39] and (Cp′₃Am)₂(μ−4,4′−bpy)[31], resulting from a decrease in ionic radii across the actinide series and is consistent with the observed M−N bond distance trend[35].

Coordination of Cp′ to curium leads to unique emission properties not before reported in a curium complex. Photoluminescence of putative Cp′₃Cm microcrystals was collected at room temperature upon irradiaton with an excitation wavelength of 420 nm (Fig. 3). A bathochromic shift and considerable splitting were observed, yielding a band centered at ~670 nm (*ca.* 14,925 cm⁻¹) with a full-width half-maximum value of about 52 nm (*ca.* 1225 cm⁻¹), noticeably larger than most previously reported[3,11,38,41,42]. While Cm³⁺ typically phosphoresces

red-orange from 590–620 nm (*ca.* 19,949–16,129 cm⁻¹)[3,11,38,41,42], red luminescence has been reported previously in CmCp₃, albiet not red-shifted to this degree[43,44]. After 24 hours of air exposure, no photoluminescence was observed. Surprisingly, 1−Cm exhibits no photoluminescence from known excitation wavelengths of 365 nm and 420 nm used for Cm³⁺, suggesting the change in coordination environment by the coordination of 4,4′−bpy to Cp′₃Cm quenches emission. We propose a possible explanation that a non-radiative deactivation mechanism attributable to the resonance between C−H vibrational modes of the 4,4′-bpy with the electronic emissive state of the curium ion (vide infra) may be responsible for this quenching.

Solid-state absorption spectroscopy of 1−Sm, 1−Gd, and 1−Cm were measured from 350–1700 nm (*ca.* 28,571–5882 cm⁻¹) (Fig. 4). Spectra of each were collected at room temperature; additionally, 1−Cm was measured at −180 °C. Assignment of the Laporte forbidden *f-f* transitions of these molecules has been completed in terms of total angular momentum, *J*. Since actinides exhibit a complex interplay between electron repulsion, relativistic, and ligand-field effects, the analysis was performed under the intermediate coupling scheme[45,46]. A charge transfer (CT) band in 1−Sm is observed beginning at 575 nm (*ca.* 17,391 cm⁻¹), masking the high energy fingerprint *f-f* transitions indicative of Sm³⁺; however, lower energy *f-f* transitions in the range of 900–1700 nm (*ca.* 11,111–5882 cm⁻¹) are still seen[47,48]. Similar to other organometallic *f*-block molecules[24,30,32], notable splitting of these transitions is detected due to the unique coordination environment resulting from the coordination of Cp′. The CT band of 1−Gd, beginning at 600 nm (*ca.* 16,667 cm⁻¹), masks the *f-f* transitions indicative of Gd³⁺ that are characteristically detected between 270–320 nm (*ca.* 37,037–31,250 cm⁻¹) (Fig. 4)[49]. Following the trend of 1−Sm and 1−Gd, the absorption spectrum of 1−Cm contains a CT band beginning at 615 nm (*ca.* 16,260 cm⁻¹) that masks high energy fingerprint transitions (Fig. 4)[10,12,18,20,44]. Transitions at 587 nm and 597 nm (*ca.* 17,036 cm⁻¹ and 16,750 cm⁻¹) (*J* = 7/2 and 5/2, respectively) typically detected at lower energy in Cm³⁺ spectra and two transitions from 630–650 nm (*ca.* 15,873–15,385 cm⁻¹) (*J* = 7/2) are observed, further displaying the unique

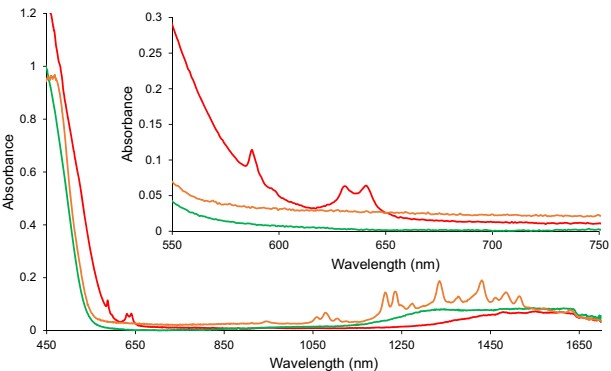

**Fig. 4 | Solid-state absorption spectra of 1−Sm, 1−Gd, and 1−Cm.** Solid-state absorption spectra of 1−Sm (orange), 1−Gd (green), and 1−Cm (red). Significant splitting of *f-f* transitions is observed in 1−Sm.

photophysical properties of 1−Cm with respect to only a small handful of solid-state Cm³⁺ compounds reported to date. These transitions are split and redshifted about 30 nm from those previously reported[42].

Solution absorption spectra of 1−Sm, 1−Gd, 1−Cm, and putative Cp′₃Cm were collected from 250–1700 nm (*ca.* 40,000–5882 cm⁻¹) at room temperature (See Supplementary Information). Characteristic *f-f* transitions of Cm³⁺ are generally very weak, possessing only single or double-digit molar absorptivities (ε), and are often difficult to observe in the solution phase[10,12,18]. As such, fingerprint transitions of Cp′₃Cm were not seen due to the low concentration. The previously stated transitions, 587 nm (*J* = 7/2), 596 nm (*J* = 5/2), and 630–650 nm (*J* = 7/2), of 1−Cm are very weak in the solution phase. However, a transition at 411 nm (*ca.* 24,331 cm⁻¹) is observed, and has undergone a bathochromic shift of about 15 nm (*ca.* 919 cm⁻¹) compared to previously reported values[12,18].

Spin-orbit CASSCF along with MC-pDFT methods (hereinafter referred to as SO-pDFT) were employed to calculate the electronic states of 1−Sm, 1−Gd, 1−Cm, and Cp′₃Cm. Since the size of the systems represents a limitation from the theoretical perspective, a model consisting of one Cp′₃M unit (M = Sm, Gd, Cm) coordinated to pyridine was used instead. The assignment of the different electronic states was done by indicating the total angular momentum quantum number *J* along with the predominant $^{2S+1}L$ term in parenthesis. The assignment of the spectroscopic transitions was performed by considering vertical excitations from the ground state (GS) to different excited states (ES).

In 1−Sm, the spin-orbit ground state corresponds to a *J* = 5/2 (⁶H). The region between -1200 cm⁻¹ (-8333 nm) to -11,150 cm⁻¹ (-897 nm) exhibit a continuum of excited states belonging to *J* = 7/2–15/2 (⁶H) and *J* = 1/2–11/2 (⁶F) manifolds. If the barycenter of the manifolds is considered, the energy gap between them is no greater than -1500 cm⁻¹. Moving to higher energies, the first excited state being predominantly quartet appears at -18,270 cm⁻¹ (-547 nm) and corresponds to a J = 5/2 (⁴G). The assignment of the experimental absorption features and the detail about the calculated SO-states is shown in Table S7, S8. As observed, theoretical predictions agree with the experimental data, particularly in the range of 11,111 cm⁻¹ (900 nm) to 5882 cm⁻¹ (1700 nm), where most of the spectroscopic features were observed. The errors associated with the calculation are in the range of 105 and 760 cm⁻¹. This has also been observed in other Sm(III) complexes, where the 4*f*–4*f* transitions were observed in the same energy region with identical assignments[48,50–52].

In 1−Gd, the scenario is simpler as it corresponds to a half-filled *f*-shell. Given its 4*f*⁷ configuration, the main transitions are expected to be in the high-energy region involving the *J* = 7/2-3/2 (⁶P) and *J* = 9/2–1/2 (⁶D) manifolds. In this case, the SO-GS corresponds to a *J* = 7/2 (⁸S) with a splitting of -15 cm⁻¹. The first excited state appears at -29,344 cm⁻¹ (341 nm) and was ascribed to the *J* = 7/2 (⁶D) manifold. The energetic

stabilization of the first excited multiplet has been previously reported for other *f*⁷ systems at different levels of theory[3,53,54]. A detailed assignment of each transition is shown in Table S9, S10. Since the CT band in 1−Gd masks the fingerprint *f-f* transitions associated with the Gd³⁺ ion, no experimental counterpart exists to assess the accuracy of the predicted transitions. Reports on other Gd(III) compounds have found the position of the first excited state to be between 32,467 cm⁻¹ (308 nm) and 31,645 cm⁻¹ (316 nm)[55–58]. The calculated value for 1−Gd exhibits a bathochromic shift of -2300 cm⁻¹ (25 nm) with respect to the range of energy found in previous reports.

Since Cm³⁺ is the isoelectronic analog of Gd³⁺, similar electronic states at different energy ranges are expected. As observed in Table S11, S12, the SO-GS of 1−Cm corresponds to a *J* = 7/2 (⁸S) with a splitting of -385 cm⁻¹. The first two excited manifolds *J* = 7/2 (⁶D) and *J* = 5/2 (⁶D), usually considered the emissive states, start to appear around 15,216 cm⁻¹ (657 nm) and 16,650 cm⁻¹ (601 nm), respectively. Therefore, the experimental absorption peaks observed between 15,873 cm⁻¹ (630 nm) and 15,384 cm⁻¹ (650 nm) can be ascribed to transitions towards these couple of excited manifolds. Regarding the Cp′₃Cm system, the assignment and position of the electronic states is almost identical to the ones found in 1−Cm. The SO-GS corresponds to a *J* = 7/2 (⁸S) exhibiting a splitting of -351 cm⁻¹, whereas the two first excited manifolds are located at -15,121 cm⁻¹ (661 nm) and 16,365 cm⁻¹ (611 nm). As shown in the experimental section (Fig. 3), the photoluminescence spectrum of this complex shows a band between -16,260 cm⁻¹ (615 nm) and -14,285 cm⁻¹ (700 nm) with three prominent peaks observed at -15,504 cm⁻¹ (645 nm), 15,152 cm⁻¹ (660 nm) and 14,925 cm⁻¹ (670 nm). This band can be assigned to transitions from the *J* = 7/2 (⁶D) manifold to the SO-GS, where the three observed peaks can be attributed to the splitting of this multiplet (Supplementary Table 12). In general, curium emits in the range of 16,949 cm⁻¹ (590 nm) to 16,129 cm⁻¹ (620 nm)[3,10,11,38,41,42]. In this case, a non-negligible bathochromic shift that was accurately reproduced by the calculations, is observed. This phenomenon has usually been attributed to the nephelauxetic effect[59] where the ligand-field of Cp′ causes a unique reduction in the inter-electronic repulsion between the *f*-electrons of curium(III).

According to SO-pDFT calculations, 1−Cm and Cp′₃Cm systems are similar in terms of assignment and positions of the SO states. Nonetheless, while 1−Cm photoluminescence is quenched, Cp′₃Cm display a prominent emission band in the vis-NIR region. It is known that C–H vibrations can lead to non-radiative routes when the low quanta superior harmonics and vibrational modes of a coordinated ligand strongly resonate with the emissive state of the metal center[60–62]. For instance, it has been reported that the fundamental vibrational modes of the C–H stretching in 4,4′−bipyridine are found at -3000 cm⁻¹[63,64]. Since C–H vibrational modes have proven to have a cumulative quenching effect with respect to the number of C–H bonds[60], we propose that a resonance between the electronic emissive state of curium(III) and the fifth vibrational harmonic of 4,4′−bipyridine to be a possible explanation for the absence of emission in 1−Cm.

Given the differences observed in the spectroscopy of curium in two similar environments, it is important to analyze the subtle differences that may arise in curium-ligand bonds when dimerizing two Cp′₃Cm units bridged by 4,4′−bpy to form 1−Cm. To do so, the natural bond orbital (NBO) approximation and the Quantum Theory of Atoms in Molecules (QTAIM) were relied upon. The analysis of the ground-state wavefunction in a localized formalism such as natural localized molecular orbitals (NLMOs) within the NBO analysis provide a unique and simple way to rationalize the chemical bond in the context of Lewis' chemical intuition. Conversely, the topological analyses of the electron density through QTAIM metrics lay out a more holistic picture of the interactions and energies associated with the electron density in the interatomic region. Optimized molecular geometry coordinates for 1−Sm, 1−Gd, 1−Cm, and Cp′₃Cm are provided as Supplementary

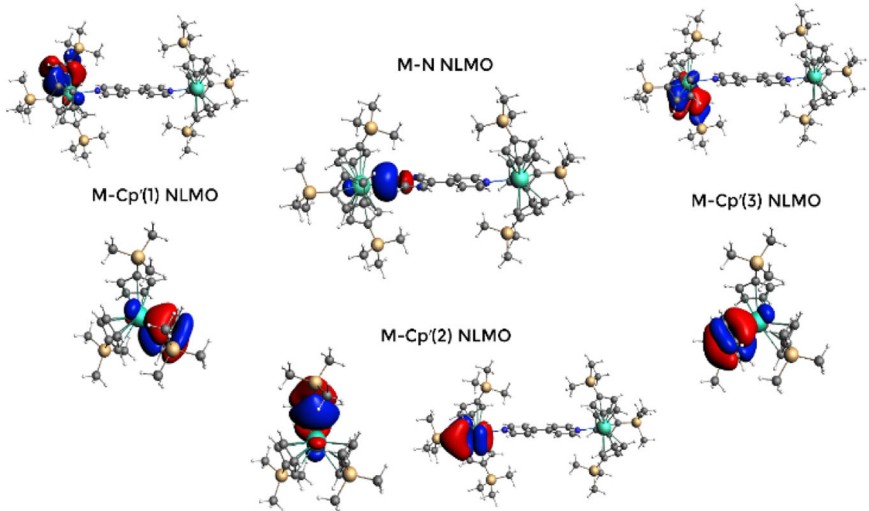

**Fig. 5 | NLMOs of 1−Cm.** Natural Localized Molecular Orbitals (NLMOs) are involved in the metal-ligand interactions occurring in 1−Sm, 1−Gd, 1−Cm, and Cp′₃Cm. The NLMOs shown in the figure correspond specifically to 1−Cm, but they are qualitatively similar to those of 1−Sm and 1−Gd.

Data 4, Supplementary Data 5, Supplementary Data 6, and Supplementary Data 7, respectively.

The nature of the actinide−Cp′ interaction is something that is still under debate because of both the formal charge and delocalization of the system, which provides a unique electronic environment compared to traditional chalcogenide-based ligands. Thus, there is a need to continue exploring the trends across the actinide series. Under the NBO localization formalism, these interactions are shown as one π-type bond with the metal center per ligand (Fig. 5). It is interesting to note that among the three analog complexes, 1−Sm displays the major engagement of $f$-orbitals to bond formation (Supplementary Table 15). This is unexpected as we generally see a greater participation of their 5$f$-electrons in bonding than the lanthanide 4$f$-orbitals. This serves as additional evidence that the cyclopentadienyl-derived ligands have unique electronic properties that result in a differential interaction between lanthanides and actinides. This scenario differs significantly when the metal−N$_{bpy}$ bonds are analyzed, where the classic trend is recovered with 1−Cm showing the stronger orbital mixing (almost twice as much the lanthanide mixing) and greater participation of the $f$-orbitals in bonding compared to 1−Sm and 1−Gd. Thus, for the metal-Cp′ bonds, 1−Sm shows an increased degree of orbital mixing with respect to 1−Cm and 1−Gd, whose metal-carbon bonds have similar hybrid contributions to their NLMOs. Whereas for metal-N$_{bpy}$ bonds 1−Sm and 1−Gd resemble each other with 1−Cm showing higher metal hybrid contributions to the bond than the lanthanide analogs (Supplementary Table 15). On the other hand, the role of the coordination of 4,4′-bpy in orbital mixing on 1−Cm is not significant as Cm−Cp′ bonds in Cp′₃Cm are almost identical to those in 1−Cm.

As a good complement to the view of the chemical bond given from localized molecular orbitals, analyzing the topology of the electron density provides a good opportunity to evaluate the accumulation of electron density, ρ(r), and the balance of kinetic and potential energies at the point where two basins (atomic fragments) contact each other along the bond path (bond critical point, BCP). Despite 1−Sm showing more pronounced Sm−Cp′ orbital mixing, the accumulation of electron density at the BCP is very similar among the three congener systems (Table 1). Conversely, the balance of potential and kinetic energy densities, V(r) and G(r), respectively, show that 1−Sm presents a similar stability of electron density at the BCP to that of 1−Cm. However, when the total energy density, H(r), is normalized against ρ(r) the difference in energy can be interpreted as effective differences in covalency. According to this metric, frequently referred to as covalency degree, it is suggested that Sm−C bonds in 1−Sm have a greater

energetic stabilization caused by covalent interactions compared to those in 1−Cm and 1−Gd by ~4 kJ mol⁻¹ and ~17 kJ mol⁻¹, respectively (Table 1). This supports the idea that the higher degree of mixing shown in Sm−Cp′ bonds correlate with the degree of covalency from an energetic perspective. On the other hand, metal−N$_{bpy}$ bonds show slightly larger ρ(r) values with Cm−N being the highest. The most striking difference is that kinetic energies associated are significantly increased, significantly increasing H(r) values and even switching to positive values as in Sm−N and Gd−N bonds. This indicates that these bonds have no covalent character and only Cm−N$_{bpy}$ bonds have a (low) covalent character. This is also reflected in the low bond orders estimated by QTAIM and NBO analyses through the delocalization index, δ(r), and Wiberg bond index (WBI) metrics, respectively. Both, δ(r) and WBI metrics show that Ln−N bonds are significantly smaller than those of the Cm−N bond. Moreover, these bonds can be compared to the previously reported values for 1−Nd and 1−Am, where both show positive values and set an unusual precedent for differences in Am(III)−N and Cm(III)−N bonds. A final comparison can be made between 1−Cm and Cp′₃Cm bonds, where from the NLMO analysis no major differences were found, but from a topological perspective we

**Table 1 | Topological metrics of the electron density of 1−Sm, 1−Gd, 1−Cm, and Cp′3Cm**

|  | 1−Sm | | 1−Gd | | 1−Cm | | Cp′₃Cm |
|---|---|---|---|---|---|---|---|
|  | Sm−C$_{avg}$ | Sm−N | Gd−C$_{avg}$ | Gd−N | Cm−C$_{avg}$ | Cm−N | Cm−C$_{avg}$ |
| ρ(r) | 0.214 | 0.238 | 0.213 | 0.244 | 0.224 | 0.275 | 0.244 |
| ∇²ρ(r) | 2.435 | 3.516 | 2.503 | 3.747 | 2.817 | 3.940 | 2.796 |
| V(r) | −486.8 | −584.6 | −491.7 | −620.4 | −556.7 | −751.6 | −611.0 |
| G(r) | 465.0 | 612.3 | 473.6 | 651.2 | 534.7 | 734.4 | 559.9 |
| H(r) | −21.8 | 27.8 | −18.1 | 30.8 | −21.9 | −17.2 | −51.0 |
| \|V(r)/G(r)\| | 1.047 | 0.955 | 1.038 | 0.953 | 1.041 | 1.023 | 1.091 |
| H(r)/ρ(r) | −101.5 | 116.4 | −85.0 | 126.3 | −98.0 | −62.6 | −209.0 |
| ε(r) | 3.797 | 0.066 | 3.713 | 0.063 | 3.766 | 0.052 | 3.349 |
| δ(r) | 0.135 | 0.180 | 0.132 | 0.183 | 0.147 | 0.227 | 0.163 |
| WBI | 0.129 | 0.161 | 0.123 | 0.168 | 0.172 | 0.243 | 0.159 |

The electron densities were derived from ground state-specific CASSCF calculations.
*Metrics tabulated correspond to values measured at the bond critical point, and Wiberg bond indices (WBI). The electron density, ρ(r) is expressed in e Å⁻³, whereas the Laplacian in e Å⁻⁵. Potential V(r), kinetic G(r), and total H(r) energy densities are expressed in kJ mol⁻¹ Å⁻³. Metal−Cp′ bonds are shown as averages of all metal−C bonds.

see significant differences. The coordination of the 4, 4′−bpy seems to reduce the accumulation of electron density at the BCP while decreasing the magnitude of the potential energy with respect to kinetic energy resulting in less than half of the total energy density in 1−Cm ($-21.9\,kJ\,mol^{-1}\,Å^{-3}$) compared to Cp′$_3$Cm ($-51.0\,kJ\,mol^{-1}\,Å^{-3}$) (Table 1). This suggests that direct correlations between orbital mixing and the degree of covalency cannot be assumed and must be contrasted carefully with alternative methods, where energies can be derived.

Characterization of (Cp′$_3$Cm)$_2$($\mu$−4,4′−bpy) and its lanthanide analogs introduces differences between An−C and Ln−C bonding. 1−Cm shares M−N similarities with its 1−Gd analog, but similar M−C bond lengths to 1−Sm, not demonstrating favor to one analog over the other and deviating from expected trends. The putative Cp′$_3$Cm complex shows lower energy emission compared traditional curium complexes. Additionally, coordination of 4,4′−bipyridine to Cp′$_3$Cm leads to a rare example of the complete quenching of curium's fingerprint emissive states. The bonding patterns calculated from these three complexes demonstrate unusual orbital mixing observed in Sm−Cp′ bonds when compared to the curium and gadolinium complexes. Overall, coordination of 4,4′−bpy does not significantly affect the orbital mixing in Cm−Cp′, but it reduces the electron density accumulated in the interatomic region that ultimately translates to lower degrees of covalency. The synthesis and characterization of (Cp′$_3$Cm)$_2$($\mu$−4,4′−bpy) leads to future studies in Cm−C bonding and necessitates further study into the unique electronic and emissive properties of this element.

## Methods

Caution! $^{248}$Cm ($t_{1/2}$ = 348,000 y) presents serious health hazards due to α-emission (5.078 MeV, 75%) and spontaneous fission (-8.4%), resulting in considerable neutron emission (80 mrem/h). All reactions and handling of $^{248}$Cm were completed in a Category II radiological facility with HEPA-equipped fume hoods and gloveboxes utilizing strict safety controls.

Anhydrous SmCl$_3$ (Sigma, 99.9%), anhydrous GdCl$_3$ (Sigma, 99.99%), 4,4′−bpy (Sigma, 98%), Bromotrimethylsilane (Sigma, 97%), Hydrobromic Acid (Sigma, 8.77 M), and distilled water were used as received. KCp′ and Cp′$_3$Sm were synthesized by their respective literature procedures[27,65]. Toluene (Sigma), hexane (Fisher Scientific), diethyl ether (Fisher Scientific), and dimethoxyethane (Sigma) were distilled using sodium benzophenone ketyl and stored on activated 3 Å molecular sieves (Sigma). Toluene, hexane, and diethyl ether were further dried over NaK for 24 hours and filtered through activated alumina neutral alumina immediately before use. Dimethoxyethane was stored on activated neutral alumina for 24 hours and filtered through more activated neutral alumina immediately before use.

All reactions were completed using Schlenk line and glovebox techniques in an argon atmosphere with exclusion of air and water unless noted otherwise. All handling of $^{248}$Cm was completed in a HEPA filter-equipped fume hood and negative pressure glovebox attached to the HEPA line.

(Cp′$_3$Sm)$_2$($\mu$−4,4′−bpy), 1−Sm. Using a scintillation vial (20 mL), the addition of 4,4′−bpy (2.6 mg, 0.017 mmol) to Cp′$_3$Sm (18.7 mg, 0.033 mmol) in toluene (2 mL) resulted in the immediate formation of an orange-yellow precipitate. The slurry was stirred vigorously overnight at room temperature, dried under reduced pressure, rinsed with hexane to remove any unreacted Cp′$_3$Sm, and once again dried under reduced pressure. The powder was dissolved in toluene and heated to 120 °C (with the cap removed) while gentle stirring, resulting in an orange solution. Upon reaching a gentle boil, stirring was ceased and the sample was manually cooled down 70 °C in 10 °C increments over a period of 20 minutes. At 70 °C the cap was reapplied to the vial carefully so as not to agitate the sample. The heat was then turned off and the vial was slowly cooled to room temperature, resulting in the

growth of yellow crystals suitable for single-crystal x-ray diffraction studies after 3 hours. UV-vis-NIR (toluene): $\lambda_{max}$ nm (cm$^{-1}$) = 944 (10,593), 1059 (9443), 1079 (9268), 1105 (9050), 1212 (8251), 1235 (8097), 1248 (8013), 1274 (7849), 1335 (7491), 1377 (7262), 1430 (6993), 1460 (6849), 1485 (6734), 1515 (6601), 1569 (6373), 1631 (6131).

(Cp′$_3$Gd)$_2$($\mu$−4,4′−bpy), 1−Gd. In a slight modification to the literature procedure[28], Cp′$_3$Gd was synthesized by the addition KCp′ (115 mg, 0.65 mmol) to anhydrous GdCl$_3$ (50 mg, 0.19 mmol) in toluene (2 mL) and stirred at 70 °C overnight. The green slurry was centrifuged and KCl was filtered off, followed by rinsing the pellet with toluene (3 × 1 mL). Toluene was evaporated under reduced pressure and the resulting powder was taken up in hexane to extract any unreacted KCp′. The slurry was again centrifuged, rinsed with hexane (3 × 1 mL), and dried under reduced pressure, resulting in a light green powder (89.4 mg, 0.157 mmol, yield 82.7%). (Cp′$_3$Gd)$_2$($\mu$−4,4′−bpy) was synthesized by the addition of 4,4′−bpy (2.5 mg, 0.016 mmol) to Cp′$_3$Gd (18.4 mg, 0.032 mmol) in toluene (2 mL). The bright yellow precipitate was dried under reduced pressure, rinsed with hexane, and taken up once again in toluene. Crystallization was completed analogous to 1−Sm, resulting in bright orange-yellow crystals suitable for single-crystal X-ray diffraction studies.

(Cp′$_3$Cm)$_2$($\mu$−4,4′−bpy), 1−Cm. A stock solution of Cm$^{3+}$ (3.3 mg Cm content, 0.013 mmol) in HCl (2 M) was dried under a nitrogen stream, dissolved in water, and transferred to a falcon tube (15 mL). Excess NH$_4$OH (-2.5 mL) was added dropwise, resulting in the immediate precipitation of Cm(OH)$_3$•nH$_2$O. The slurry was centrifuged and the pellet was rinsed with water (3 × 1 mL). Cm(OH)$_3$•nH$_2$O was dissolved in minimal HBr (1 mL, 8.77 M), pipetted to a scintillation vial (20 mL), and dried under a stream of nitrogen. The colorless CmBr$_3$•nH$_2$O was rinsed with diethyl ether (2 × 1 mL) and transported into a glovebox overnight.

Following actinide drying procedures reported previously[30,32,66], DME (1.5 mL) was added dropwise to CmBr$_3$•nH$_2$O. The slurry was stirred for 10 minutes, followed by the dropwise addition of bromotrimethylsilane (TMS−Br, 1.5 mL). The slurry was stirred at 50 °C for 2 hours and the powder began to dissolve slowly. After cooling to room temperature, hexane (4 mL) was added, resulting in a white precipitate. The sample was allowed to settle for 15 minutes before the supernatant was pipetted away, and then was rinsed with hexane (3 × 2 mL), stirring for 5 minutes and resting for 15 minutes between each rinse, followed by drying under reduced pressure for 30 minutes. The resulting white powder was taken up in diethyl ether (1.5 mL), stirred for 20 minutes, and further precipitated out by the addition of hexane (4 mL). The slurry was stirred vigorously for 5 minutes and allowed to settle for 15 minutes before pipetting away the supernatant and drying under reduced pressure for 2 hours, resulting in a white powder of putative CmBr$_3$(DME)$_n$.

KCp′ (8.2 mg, 0.046 mmol) was dissolved in toluene (1.5 mL), added dropwise to CmBr$_3$(DME)$_n$, and stirred vigorously at 70 °C for 2 hours. A color change from colorless to tan was observed. The sample was centrifuged to remove the KBr byproduct and filtered, followed by rinsing the pellets with toluene (3 × 0.5 mL each pellet). Toluene was evaporated under reduced pressure and hexane (1.5 mL) was added to precipitate and extract excess KCp′. The slurry was centrifuged, supernatant filtered, and the pellets were washed with hexane (3 × 0.5 mL each pellet). About 40 µL of the resulting champagne-colored solution was isolated and diluted to 1.5 mL for solution phase absorption studies. Putative CmCp′$_3$ was isolated and dried under reduced pressure, resulting in a tan oil (7.1 mg, 0.011 mmol, yield 81%). Two crystal clusters were extracted for photoluminescence studies. The crystal quality was not suitable for single-crystal X-ray diffraction but were suitable for emission studies.

An immediate color change to bright yellow was observed upon the addition of 4,4′−bpy (1.0 mg, 0.006 mmol) to CmCp′$_3$ in toluene (2 mL). The solution was transferred from a 20 mL scintillation vial to a 6 mL scintillation vial and concentrated under reduced pressure until

an orange-yellow powder was observed (~0.3 mL toluene). The slurry was slowly stirred and heated to 120 °C resulting in the orange powder dissolving. Similar to 1−Sm and 1−Gd, the solution was slowly cooled to 70 °C in 10 °C increments. At 70 °C the vial was capped and the hot-plate was turned off, allowing the sample to slowly cool to room temperature. Around 60 °C a color change from orange to an emerald green was observed, followed by a slow color change to a sapphire blue over a period of about an hour. The blue solution rested overnight at room temperature. The following morning the sample was again yellow and gold crystals suitable for single-crystal X-ray diffraction were retrieved. UV-vis-NIR (toluene): $\lambda_{max}$ nm (cm$^{-1}$) = 587 (17,036), 597 (16,750), 631 (15,848), 641 (15,601).

## Data availability

The structural data generated in this study have been deposited in the Cambridge Structural Database (CSD) under accession codes 2236796, 2236795, and 2236794 for 1−Sm, 1−Gd, and 1−Cm, respectively. The spectroscopic, crystallographic, and theoretical data for this study are provided in the Supplementary Information. Raw spectroscopic data generated in this study is provided in the Source Data file. Cif files for 1−Sm, 1−Gd, and 1−Cm structural data are provided as Supplementary Data 1−3, respectively. Optimized molecular geometry xyz coordinates for 1−Sm, 1−Gd, 1−Cm, and Cp′$_3$Cm are provided as Supplementary Data 4−7, respectively. All other data are available from the corresponding author upon request. Source data are provided with this paper.

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

## Acknowledgements

This study was completed under the Department of Energy Office of Science, Basic Energy Sciences, DOE-BES Heavy Elements Chemistry award number DE-FG02-13ER16414 (TEAS). We would like to thank Oak Ridge National Laboratory for providing the $^{248}$Cm. Additionally, we would like to extend our gratitude to Professor Cory J. Windorff for his help and insight into organometallic actinide chemistry, as well as our radiation safety officers, Ashley Gray and Jason Johnson, for their efforts in our program.

## Author contributions

B.N.L. completed synthetic work, single-crystal X-ray diffraction, solid-state UV-vis, and solution phase UV-vis. M.B.L. completed CASSCF and spectroscopic calculations. J.M.S. assisted with synthetic work, completed solid-state UV-vis, and collected photoluminscence. T.N.P. completed low-temperature photoluminscence and spectroscopy. C.C.-B. completed bonding calculations and assisted in overseeing the project. T.E.A.S. was project overseer and provided the materials.

## Competing interests

The authors declare no competing interests.
