## [Peer Review File · Nature Communications]

Altering the Spectroscopy, Electronic Structure, and Bonding of Organometallic Curium(III) Upon Coordination of 4,4'-bipyridineReviewers' Comments:

Reviewer #1:

Remarks to the Author:

In the manuscript entitled 'Altering the Spectroscopy, Electronic Structure, and Bonding of Organometallic Curium(III) Upon Coordination of 4,4'-bipyridine' by Thomas Albrecht-Schönzart and his group a rare example is described of an organometallic Curium (Cm) complex, (CpTMS)₃Cm4,4'-bipyridine, resulting from the coordination of 4,4'-bipyridine to (CpTMS)₃Cm plus the effects on electronic properties while coordination. The manuscript contains the first structurally characterised organometallic Cm complex with pi ligands.

Generally, it is a high quality work, the procedures described and the documentation provided are very well. The interpretation of the data, figures also is all very well. The paper should be published in Nat Comm.

I would encourage the authors to extend the theoretical modelling in order to understand better general trends in structure and electronics, but I think this is not to be done within this manuscript.

Single points:

Introduction: the Cm problem in the radioactivity of the nuclear waste is not known only since 2016 (lit 5) or 2019 (lit 6), a lot of research is done in this area and I would ask the authors to introduce a more general and recent review or book citation on this. I will not provide an example.

In the cited literature 26 to 29 one citation could be added on PuCp₃ and its structural characterisation

Results and discussion:

In the complexes (CpTMS)₃An4,4'-bipyridine (An = U, Am, Cm) a shortening of the An-N bond is observed from 2.626(7) Å, 2.618(2) Å to 2.5962(16) Å, the authors correlate this to a trend across the actinide series but do not link it to actinide contraction. Why is the trend going from U to Am weaker expressed than while going from Am to Cm? The trend is, however, parallel to the decrease in the mean M-Ct distance from 2.540(10) over 2.524(3) to 2.517(2) Å in the series U, Am, Cm but here it is quite linear... (ca 0.005 Å per Z). Do the authors have an explanation for this?

The M-C distances normally vary over a broader range which is reflected in the reported data in S4, I calculate the for example for Am a mean of 2.7952 Å but with a sigma of 0.044 Å. Please check the standard deviations given in table S4 for the M-Cavg bond length. In the structure discussion the authors focus a lot on comparing their lanthanides analogues to (CpTMS)₃Cm4,4'-bipyridine. This is all based on the Shannon radii published in 1976. Generally I would encourage to compare the lanthanides to their lanthanides analogues first and to look for trend, then doing the same for the actinide, finally confronting the two trends looking for similarity and differences...

The authors have performed as well DFT calculations on the system. I wonder why not more structures have been calculated and included, so for Cp₃An an extension towards Np would have enabled to embed the experimental data of Np in the comparison as well for Pu in the case of the Cp(TMS)₂ ligands... so for example the mean Np-Ccenter in Cp₃Np is 2.482 Å. It could have been interesting.

I disagree with the authors statement 'The nature of the metal-Cp' interaction is something that is not well-understood because of the delocalization on the Cp' ring.' This is too general. The nature of the bonding is understood very well understood for most of the metals and well enough for the actinides but we still need to explore its trends over the actinide series with the actinides having different frontier orbitals with comparable energies which could finally contribute to the bonding.

I am not sure whether the reason for the quenching of the photoluminescence of (CpTMS)₃Cm4,4'-bipyridine is really an effect of a resonance between the electronic emissive state of curium(III) and the fifth vibrational harmonic of 4,4'-bipyridine. In order to prove this a deuterated ligand should be used or it should be correct as well for other ligands with aromatic CH bonds... such as the simple pyridyl... This evidence is not given here, of course. Therefore, I would ask the authors to think about to weaken down this pathway as a possible explanation of the quench but that more extended investigations are needed in order to really understand the quenching mechanism.

The conclusion will depending on the changes the authors do need some attention, but I would recommend focusing on Cm and its unique and surprising properties, which ask for more detailed

investigation in the future.

Reviewer #2:

Remarks to the Author:

This paper reports on the synthesis of Cm, Gd and Sm silylated cyclopentadienyl complexes and their interaction with 4,4'-bipyridine. The structures of each complex are presented and comparisons are made between them. The work on Cm is pivotal to the understanding of Cm-C interactions, and a search of the CCDC shows that while CmCp₃ has been structurally evaluated, only powder data were available and showed it was isomorphous with the praseodymium analogue. Therefore, bonding parameters could not be easily inferred from that data. The current study provides such information. The coordination of 4,4'-bipyridine led to quenching of the compounds emissive states and the paper details this phenomenon, while theoretical studies have supported observations found.

This study is a rare example of organometallic chemistry of transuranics and while access to these elements is not trivial, meaning it is only available to some research groups, the study provides much information on the chemistry of Cm.

I found the paper easily readable and the work very detailed, and this exploratory work on Cm is certainly the type of work that should be of interest to readers of this journal.

I support publication of this paper in its present form as the compounds are well characterised and the work nicely done.

Reviewer #3:

Remarks to the Author:

First of all, based on my expertise, I express here my evaluation of the paper from a general point of view, focusing on the quantum chemical calculations applied to f-complexes and their link with the experiments.

This manuscript from Albrecht-Schönzart et al. presents a combined experimental and theoretical investigation of 4f and 5f organometallic complexes (Sm, Gd, Cm) coordinated to the 4,4'-bipyridine ligand. It is a good quality piece of work. The article is well written. The description of the methodology and the presentation of the results are clear. The main conclusions are that both experimentally and theoretically, a partial bond between an f-element and a carbon atom has been established, and that there is a strong difference in the emissive state between the Cm complex complexed or not to bipyridine.

This paper will bring new perspectives in the field of f-element chemistry. I don't see any real issue with the publication, but I think the conclusions need to be further strengthened by the analysis of the results. Therefore, I have a few comments for the authors below.

1) Introduction and references. There is room to add some more discussion and dedicated references to the organometallic chemistry of lanthanides. In particular, the introduction mentions the role of the lanthanide complexes as a reference or benchmark but limited information is provided in the text.

2) If I read the supplementary material correctly, the DFT calculations do not account for dispersion effects. The DFT-D3 method (as an example) is now commonly used and it has been reported several times that there is an effect of dispersion forces on the geometries of such complexes, especially those containing Cp ligands and their derivatives. Some changes in geometries inevitably lead to changes in chemical bonding. Can the authors comment on this point?

3) I strongly appreciate the authors' approach of using the complementarity between NBO and QTAIM analyses to understand the nature of metal-ligand interactions. The results indicate a weak covalency for the Cm-N(bpy) bond. I think that an ELF analysis could confirm or not this result. ELF analysis

would also provide information on M-Cp' bonds.

4)Figure 5 is not easy to read mainly because of the size of the images (and the resolution). What are the complexes presented here? Are there any differences between the Sm, Gd and Cm complexes? Why is it necessary to present 3 NLMOs for M-Cp'(1, 2, 3)? The caption should be more detailed.

5)To finish, I find that the conclusions are a little hasty and do not sufficiently emphasize the significance of the new results obtained on Cm complexes and the specificities of these complexes compared to those of lanthanides.

Minor points

6)Introduction, page 5 : The Cm-N distances of 1-Cm and Cm(S₂CNEt₂)₃(N₂C₁₂H₈) are very similar. I suggest avoiding the term "less".

7)Page 5: Th⁴⁺ complex. Could the authors provide the value of the distance to allow a direct comparison?

8)Wiber bond indices are provided in Table 1 but not commented in the text. Could the author extract information from these indices?

9)I would recommend providing the xyz coordinate files of the complexes as independent text files for easy reuse instead of inserting them into the supplementary material.

Response to reviewers

We would like to thank the reviewers for their suggestions, each of which has greatly improved the quality of the manuscript. Below are the specific responses to reviewer comments and how they were addressed in the manuscript:

Reviewer #1 (Remarks to the Author):

In the manuscript entitled ‘Altering the Spectroscopy, Electronic Structure, and Bonding of Organometallic Curium(III) Upon Coordination of 4,4’-bipyridine’ by Thomas Albrecht-Schönzart and his group a rare example is described of an organometallic Curium (Cm) complex, (CpTMS)₃Cm4,4’-bipyridine, resulting from the coordination of 4,4’-bipyridine to (CpTMS)₃Cm plus the effects on electronic properties while coordination. The manuscript contains the first structurally characterised organometallic Cm complex with pi ligands. Generally, it is a high quality work, the procedures described and the documentation provided are very well. The interpretation of the data, figures aso is all very well. The paper should be published in Nat Comm.

Response: We would like to thank the reviewer for the supportive and thought-provoking feedback on the manuscript and have addressed suggestions below.

I would encourage the authors to extend the theoretical modelling in order to understand better general trends in structure and electronics, but I think this is not to be done within this manuscript.

Response: We thank the reviewer for the comment. This is something we are in the process of doing with similar systems.

Single points:

Introduction: the Cm problem in the radioactivity of the nuclear waste is not known only since 2016 (lit 5) or 2019 (lit 6), a lot of research is done in this area and I would ask the authors to introduce a more general and recent review or book citation on this. I will not provide an example.

Response: Additional references ranging over a larger time period have been added to support these statements.

In the cited literature 26 to 29 one citation could be added on PuCp₃ and its structural characterization

Response: References to Pu-C bonding have been added to the suggested location.

Results and discussion:

In the complexes (CpTMS)₃An_{4,4'}-bipyridine (An = U, Am, Cm) a shortening of the An-N bond is observed from 2.626(7) Å, 2.618(2) Å to 2.5962(16) Å, the authors correlate this to a trend across the actinide series but do not link it to actinide contraction. Why is the trend going from U to Am weaker expressed than while going from Am to Cm? The trend is, however, parallel to the decrease in the mean M-Ct distance from 2.540(10) over 2.524(3) to 2.517(2) Å in the series U, Am, Cm but here it is quite linear... (ca 0.005 Å per Z). Do the authors have an explanation for this?

Response: We believe a possible explanation could be due to the covalency observed in U-N and Cm-N bonding, but the Am-N nitrogen bond was reported as ionic per the QTAIM studies, leading Am to be an outlier in this situation instead of Cm. We have added a statement regarding this to the structural section, presenting it as a hypothesis since we do not have a definitive, concrete explanation for this unusual trend.

The M-C distances normally vary over a broader range which is reflected in the reported data in S4, I calculate the for example for Am a mean of 2.7952 Å but with a sigma of 0.044 Å. Please check the standard deviations given in table S4 for the M-Cavg bond length.

Response: Thank you for catching this. The standard deviations in the table were calculated incorrectly. The standard deviations have been corrected and are updated in both the table and the main manuscript.

In the structure discussion the authors focus a lot on comparing their lanthanides analogues to (CpTMS)₃Cm_{4,4'}-bipyridine. This is all based on the Shannon radii published in 1976. Generally I would encourage to compare the lanthanides to their lanthanides analogues first and to look for

trend, then doing the same for the actinide, finally confronting the two trends looking for similarity and differences...

Response: A brief lanthanide-lanthanide comparison of M-N distances has been added before the Cm-lanthanide comparisons to help the discussion read smoother, in addition to some comparison in the M-Cent and M-C sections.

The authors have performed as well DFT calculations on the system. I wonder why not more structures have been calculated and included, so for Cp'3An an extension towards Np would have enabled to embed the experimental data of Np in the comparison as well for Pu in the case of the Cp(TMS)2 ligands... so for example the mean Np-Ccenter in Cp'3Np is 2.482 Å. It could have been interesting.

Response: We have decided to stick with the experimental structures only as performing the full analyses on more other actinide systems would extend too much the discussion and deviate the attention from Cm that is the main focus of this manuscript. We do believe that extending the analysis through Np would help finding trends in electronic structure from a bonding perspective.

I disagree with the authors statement 'The nature of the metal-Cp' interaction is something that is not well-understood because of the delocalization on the Cp' ring.' This is too general. The nature of the bonding is understood very well understood for most of the metals and well enough for the actinides but we still need to explore its trends over the actinide series with the actinides having different frontier orbitals with comparable energies which could finally contribute to the bonding.

Response: The statement has been modified to "The nature of the actinide-Cp' interaction is something that is still under debate because of both the formal charge and delocalization of the system, which provides a unique electronic environment compared to traditional chalcogenide-based ligands. Thus, there is a need for continue exploring the trends across the actinide series."

I am not sure whether the reason for the quenching of the photoluminescence of (CpTMS)3Cm4,4'-bipyridine is really an effect of a resonance between the electronic emissive state of curium(III) and the fifth vibrational harmonic of 4,4'-bipyridine. In order to prove this a deuterated ligand should be used or it should be correct as well for other ligands with aromatic CH bonds... such as the simple pyridyl.... This evidence is not given here, of course. Therefor, I would

ask the authors to think about to weaken down this pathway as a possible explanation of the quench but that more extended investigations are needed in order to really understand the quenching mechanism.

Response: We thank the reviewer for pointing this out. The quenching of the photoluminescence is only a viable explanation. We have toned down the conclusions and the main text regarding the quenching.

The conclusion will depending on the changings the authors do need some attention, but I would recommend focusing on Cm and its unique and surprising properties, which ask for more detailed investigation in the future.

Response: The conclusion has been modified to have a stronger focus on the unique Cm properties as suggested.

Reviewer #2 (Remarks to the Author):

This paper reports on the synthesis of Cm, Gd and Sm silylated cyclopentadienyl complexes and their interaction with 4,4'-bipyridine. The structures of each complex are presented and comparisons are made between them. The work on Cm is pivotal to the understanding of Cm-C interactions, and a search of the CCDC shows that while CmCp3 has been structurally evaluated, only powder data were available and showed it was isomorphous with the praseodymium analogue. Therefore, bonding parameters could not be easily inferred form that data. The current study provides such information. The coordination of 4,4'bipyridne led to quenching of the compounds emissive states and the paper details this phenomenon , while theoretical studies have supported observations found.

This study is a rare example of organometallic chemistry of transuranics and while access to these elements is not trivial, meaning it is only available to some research groups, the study provides much information on the chemistry of Cm.

I found the paper easily readable and the work very detailed, and this exploratory work on Cm is certainly the type of work that should be of interest to readers of this journal.

I support publication of this paper in its present form as the compounds are well characterised and the work nicely done.

Response: We would like to thank the reviewer for reviewing the manuscript and support of this manuscript. The kind comments are greatly appreciated.

Reviewer #3 (Remarks to the Author):

First of all, based on my expertise, I express here my evaluation of the paper from a general point of view, focusing on the quantum chemical calculations applied to f-complexes and their link with the experiments.

This manuscript from Albrecht-Schönzart et al. presents a combined experimental and theoretical investigation of 4f and 5f organometallic complexes (Sm, Gd, Cm) coordinated to the 4,4'-bipyridine ligand. It is a good quality piece of work. The article is well written. The description of the methodology and the presentation of the results are clear. The main conclusions are that both experimentally and theoretically, a partial bond between an f-element and a carbon atom has been established, and that there is a strong difference in the emissive state between the Cm complex complexed or not to bipyridine.

This paper will bring new perspectives in the field of f-element chemistry. I don't see any real issue with the publication, but I think the conclusions need to be further strengthened by the analysis of the results. Therefore, I have a few comments for the authors below.

Response: We would like to thank the reviewer for the supportive and thought-provoking feedback on the manuscript and have addressed suggestions below.

1) Introduction and references. There is room to add some more discussion and dedicated references to the organometallic chemistry of lanthanides. In particular, the introduction mentions the role of the lanthanide complexes as a reference or benchmark but limited information is provided in the text.

Response: Discussion on the contribution of lanthanide organometallic chemistry and how we apply it to the mid-actinides has been added to the introduction.

2) If I read the supplementary material correctly, the DFT calculations do not account for dispersion effects. The DFT-D3 method (as an example) is now commonly used and it has been reported several times that there is an effect of dispersion forces on the geometries of such complexes, especially those containing Cp ligands and their derivatives. Some changes in geometries inevitably lead to changes in chemical bonding. Can the authors comment on this point?

Response: We partially agree with the reviewer. Ideally, we include dispersion corrections to geometry optimizations for lanthanides and actinides up to uranium. However, the fact that these dispersion corrections have not been tested for trans-uranic complexes makes those results less reliable. In fact, ADF does not include dispersion correction for atoms with $Z > 94$ (Pu), this means that the pair-wise interaction between Cm-C(Cp') are not tested and the atomic dispersion not included. However, we still tested the results for CmCp'3, where the RMSD for the full structure was 0.180 Å, but when the trimethylsilyl groups were removed the RMSD decreased to 0.073 Å (see figures below). Finally, we ran NBO calculations that confirm that the nature of the bond does not change with such small changes in the geometry.

3) I strongly appreciate the authors' approach of using the complementarity between NBO and QTAIM analyses to understand the nature of metal-ligand interactions. The results indicate a weak

covalency for the Cm-N(bpy) bond. I think that an ELF analysis could confirm or not this result. ELF analysis would also provide information on M-Cp' bonds.

Response: We appreciate the suggestion from the reviewer and we think it would be interesting to perform some ELF studies on systems like these in the future. However, we think that the analyses included provide sufficient evidence of the nature of the chemical bond and that further analyses would be more appropriate for more specific computationally oriented manuscripts.

4)Figure 5 is not easy to read mainly because of the size of the images (and the resolution). What are the complexes presented here? Are there any differences between the Sm, Gd and Cm complexes? Why is it necessary to present 3 NLMOs for M-Cp'(1, 2, 3)? The caption should be more detailed.

Response: The figures provided and uploaded are high resolution. It is possible that the conversion to PDF files might have reduced the quality of it. The caption has been edited with more details of the NLMOs. The intent of the picture is to show that the orbitals shown in molecular complexes 1-M are qualitatively almost identical with those observed in the MCp'₃ complexes. With respect to why it is necessary to present 3 NLMOs for the M-Cp' interactions is because those are the only ones that result from an NBO analysis, which has been previously discussed in a paper from our group on Pu, U, and Ce Cp' complexes (C Celis-Barros, T Albrecht-Schonzart, CJ Windorff. Organometallics 40 (11), 1577-1587).

5)To finish, I find that the conclusions are a little hasty and do not sufficiently emphasize the significance of the new results obtained on Cm complexes and the specificities of these complexes compared to those of lanthanides.

Response: The conclusions have been updated to highlight the important properties of the Cm complex.

Minor points

6)Introduction, page 5 : The Cm-N distances of 1-Cm and Cm(S₂CNEt₂)₃(N₂C₁₂H₈) are very similar. I suggest avoiding the term "less".

Response: Wording has been changed to express similarity in this comparison.

7)Page 5: Th⁴⁺ complex. Could the authors provide the value of the distance to allow a direct comparison?

Response: The M-N value for Th⁴⁺ and a comparison to the trivalent systems has been added.

8)Wiber bond indices are provided in Table 1 but not commented in the text. Could the author extract information from these indices?

Response: We have added one additional sentence showing that delocalization indices and WBIs provide the same qualitative picture about the M-C and M-N bond orders.

9)I would recommend providing the xyz coordinate files of the complexes as independent text files for easy reuse instead of inserting them into the supplementary material.

Response: We have provided the xyz coordinate files as additional text files as suggested.

Reviewers' Comments:

Reviewer #1:

Remarks to the Author:

Dear authors,

I am content with the changes provided in the manuscript and recommend publication as is.

KR

Reviewer #2:

Remarks to the Author:

I am satisfied with the revisions

Reviewer #3:

Remarks to the Author:

The authors have answered my comments and questions. I appreciate the effort to provide additional results on the effect of dispersion on geometries in the answer. This result is interesting.

I support the publication of this manuscript in its present form.

Response to reviewers

Reviewer #1 (Remarks to the Author):

Dear authors,

I am content with the changes provided in the manuscript and recommend publication as is.

KR

Response: We would like to thank the reviewer for their feedback and assistance in improving the manuscript.

Reviewer #2 (Remarks to the Author):

I am satisfied with the revisions

Response: We would like to thank the reviewer for their feedback and support for the manuscript.

Reviewer #3 (Remarks to the Author):

The authors have answered my comments and questions. I appreciate the effort to provide additional results on the effect of dispersion on geometries in the answer. This result is interesting.

I support the publication of this manuscript in its present form.

Response: We would like to thank the reviewer for their feedback and assistance on improving the theoretical portion of the manuscript.